



# Object-based analysis of simulated thunderstorms in Switzerland: application and validation of automated thunderstorm tracking on simulation data

Timothy H. Raupach[1,2,5], Andrey Martynov[1,2], Luca Nisi[4], Alessandro Hering[4], Yannick Barton[1,2], and Olivia Martius[1,2,3]

[1]Oeschger Centre for Climate Change Research, University of Bern, Bern, Switzerland
[2]Institute of Geography, University of Bern, Bern, Switzerland
[3]Mobiliar Laboratory for Natural Risks, University of Bern, Bern, Switzerland
[4]Federal Office of Climatology and Meteorology MeteoSwiss, Locarno, Switzerland
[5]Present address: Climate Change Research Centre, University of New South Wales, Sydney, Australia

**Correspondence:** T. H. Raupach (timothy.h.raupach@gmail.com)

**Abstract.** We present a feasibility study for an object-based method to characterise thunderstorm properties in simulation data from convection-permitting weather models. An existing thunderstorm tracker, the Thunderstorm Identification, Tracking, Analysis and Nowcasting (TITAN) algorithm, was applied to thunderstorms simulated by the Advanced Research Weather Research and Forecasting (AR-WRF) weather model at convection-permitting resolution for a domain centred on Switzerland.

Three WRF microphysics parameterisations were tested. The results are compared to independent radar-based observations of thunderstorms derived using the MeteoSwiss Thunderstorms Radar Tracking (TRT) algorithm. TRT was specifically designed to track thunderstorms over the complex Alpine topography of Switzerland. The object-based approach produces statistics on the simulated thunderstorms that can be compared to object-based observation data. The results indicate that the simulations underestimated the occurrence of severe and very large hail compared to the observations. Other properties, including the

number of storm cells per day, geographical storm hotspots, thunderstorm diurnal cycles, and storm movement directions and velocities, provide a reasonable match to the observations, which shows the feasibility of the technique for characterisation of simulated thunderstorms over complex terrain.

## 1 Introduction

Convection-permitting simulations will play a critical role in reducing the existing high uncertainty around the responses of

thunderstorms (e.g. Diffenbaugh et al., 2013; Collins et al., 2013; Hartmann et al., 2013; Allen, 2018), and hailstorms (e.g. Martius et al., 2018; Allen, 2018; Raupach et al., 2021b) to climate change. Such models have sufficiently high resolution to explicitly resolve individual storm structures without parameterised convection (e.g. Weisman et al., 1997; Bryan et al., 2003), and thus address thunderstorm initiation, which cannot easily be addressed if proxy relationships are used to infer information about thunderstorm environments (e.g. Tippett et al., 2015). High-resolution simulations can be difficult to compare either to

one another or to observations, since mismatches in timing or location of weather features can occur even when the overall





statistical properties of the weather phenomena are in agreement, leading to point-to-point comparisons results that do not properly show model performance (e.g. Ebert, 2009; Gilleland et al., 2010). Object- or feature-based comparisons are one way to address this problem (e.g. Ebert, 2009; Gilleland et al., 2010). In the object-based approach, objects – storm cells, for example – are identified individually and their number and properties calculated and compared. Object-based approaches have been used to verify forecasts from numerical weather models (e.g. Done et al., 2004; Davis et al., 2006a, b), including through the use of storm-cell tracking methods (Pinto et al., 2007; Caine et al., 2013), and they are a useful way to statistically summarise and compare model outputs and observations that may be otherwise difficult to compare (e.g. Gilleland et al., 2010; Caine et al., 2013). In this article we present a feasibility study to investigate the ability of an existing radar-based thunderstorm tracker to perform object-based analysis of simulated thunderstorms in the topographically complex region of Switzerland.

Simulations were run using the Advanced Research Weather Research and Forecasting (AR-WRF, version 4.0.1, hereafter WRF) weather model (Skamarock et al., 2019) at convection-permitting grid spacing and high temporal resolution for the month of May 2018. Thunderstorms were identified in the model output using the Thunderstorm Identification, Tracking, Analysis and Nowcasting (TITAN, Dixon and Wiener, 1993) algorithm (git version lrose-cyclone-20190801-167-g85b01e9a3) run on simulated radar reflectivity fields. In this paper the results are compared to a database of thunderstorm observations for Switzerland (Nisi et al., 2018). These observations were made using the Swiss radar network and the MeteoSwiss-developed Thunderstorms Radar Tracking (TRT, Hering et al., 2004, 2008) algorithm. We consider TRT results to be representative of the thunderstorm environment in Switzerland, and tested simulated thunderstorm results against this benchmark.

TITAN has previously been applied to WRF output: Pinto et al. (2007) used TITAN on WRF simulations and corresponding radar observations in the southeastern United States of America, and found that although the WRF simulations produced storms that initiated at similar times as the observed storms, there were differences between the modelled and observed storm evolution and spatial coverage. More recently, Caine et al. (2013) used TITAN to compare WRF output and radar data for tropical storms in northern Australia. They showed the advantages of an object-based approach for comparing models to observations, and used it to identify that WRF produced overly tall and small convective cells. Our study is the first to apply such a technique to the complex Alpine domain of Switzerland. A difference from previous studies is that we compare simulated thunderstorm properties to radar observations characterised by an independent thunderstorm tracker designed specifically for the Swiss domain, thus testing the ability of WRF and TITAN to characterise thunderstorms in the challenging Alpine environment.

In this work we aim to answer the question of whether storm properties produced using WRF and TITAN are reasonably representative of storms observed in Switzerland. If this question is answered in the positive, then this processing approach provides a useful way to study future severe storm scenarios for Switzerland and other complex domains. The rest of this article is organised as follows: the data and methods used are described in Section 2. Results of the simulation to observation comparisons are shown in Section 3. Implications of the results are discussed and conclusions are drawn in Section 4.





## 2   Data and methods

In this section we introduce the data and methods used in this work, starting with the study time period and location in Section
2.1. The reference data set, which is used as the ground-truth for storm characterisation, is introduced in Section 2.2. The
TITAN storm tracker is described in Section 2.3. The weather model we used to simulate thunderstorms is described in Section
2.4. The methods by which the storm properties for simulations are compared to the reference data set are explained in Section
2.5. Finally, optimisation of TITAN threshold parameters is described in Section 2.6.

### 2.1   Study period and domain

The study domain is centred over Switzerland, an area in which complex topography affects precipitation processes (e.g.
Houze Jr., 2012) and our ability to monitor them (e.g. Germann et al., 2015; Speirs et al., 2017). Thunderstorms (van Delden,
2001) and hailstorms (Houze et al., 1993; Willemse, 1995; Punge and Kunz, 2016; Nisi et al., 2016; Punge et al., 2017;
Madonna et al., 2018) are a regular warm-season occurrences in Switzerland. The Swiss convective season runs from April to
September, with storms occuring primarily in the foothill regions north and south of the main Alpine range, and in the Jura
mountains (Nisi et al., 2016, 2018). The most populous area in Switzerland – the Swiss Plateau, between the Jura and the Alps
– is regularly affected by severe thunderstorms that can be long-lived and produce hail (Houze et al., 1993; Nisi et al., 2018).
Historical cases include storms that inflicted significant damage (e.g. Schmid et al., 1997, 2000; Peyraud, 2013; Trefalt et al.,
2018). In Switzerland, severe storms are monitored primarily by a dual-polarisation radar network operated by MeteoSwiss
(Germann et al., 2015). Switzerland's climate is expected to be significantly affected by global warming (CH2018, 2018), but
there remains high uncertainty on the likely future evolution of severe thunderstorms in Switzerland (CH2018, 2018; Willemse,
70   1995).

Figures 1 and 2 show the geographical area of the study, with the radar coverage area overlaid. The Alps run across the
centre of the simulation domain and split it into northern and southern regions. Figure 3 shows the sub-domains used in this
study; these correspond to geographical features and are modified versions of the domains used by Nisi et al. (2016). Table 1
lists the coordinates of the boundaries of the study domain, which was chosen to be well covered by both the radar data and
simulations.

The study period was May 2018. In Switzerland, the 2018 convective season was characterised by lower than average overall
rainfall (MétéoSuisse, 2018c), but high levels of convective activity in late May and early June (MétéoSuisse, 2018b, a). In
May, thunderstorms occurred in Switzerland on days 6–9 and 11–13 of the month, and then almost daily from the 15th until
the end of the month (MétéoSuisse, 2018b). 22 May saw thunderstorms across the Central Plateau with a 30-year daily rain
amount (73.2 mm) at Belp, and on 30 and 31 May there were extensive hailstorms over the Swiss Plateau that caused local
flooding (MétéoSuisse, 2018b). Hail was reported in Switzerland on, 7, 8, 15, 21, 30, and 31 May (Sturmarchiv Schweiz).





**Table 1.** Corner point coordinates for the study domain. E and N are the Swiss coordinates (CH1903+/LV95) in the east and north directions respectively, while Lon and Lat are the corresponding longitude and latitude. L and R stand for left and right respectively.

| Corner | E [m] | N [m] | Lon [°] | Lat [°] |
|---|---|---|---|---|
| Bottom L | 2464500 | 1056000 | 5.70093 | 45.64222 |
| Top L | 2464500 | 1316000 | 5.62372 | 47.98025 |
| Top R | 2854500 | 1316000 | 10.84586 | 47.94466 |
| Bottom R | 2854500 | 1056000 | 10.70117 | 45.60812 |

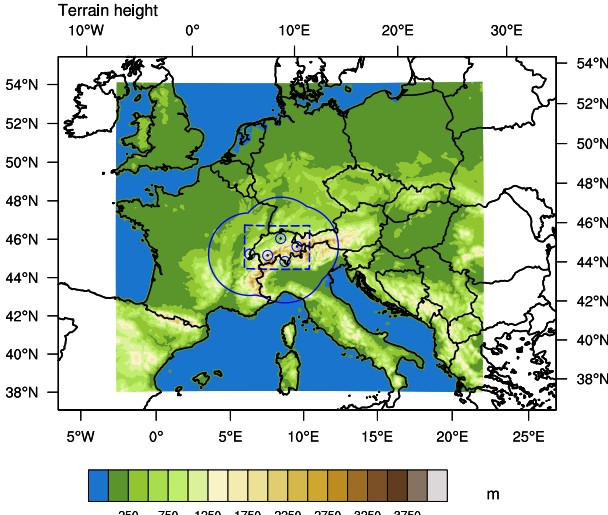

**Figure 1.** Terrain heights (above sea level) for points covered by the WRF simulation outer domain. Black lines show national borders and coastlines. The locations of the five MeteoSwiss radars are indicated with blue circled points, and the solid blue line shows the approximate radar domain. The dashed blue line shows the study domain. Storms with centre points outside the study domain are not considered in this study. Elevations below 0.001 m are plotted in blue. Plot produced using NCL version 6.6.2.

## 2.2 Reference thunderstorm data set

The reference data for thunderstorms in Switzerland is a database of thunderstorm tracking results compiled by MeteoSwiss. MeteoSwiss operates five C-band, dual-polarisation, Doppler weather radars in a network designed for high performance 85 despite the challenges posed by the mountainous terrain of Switzerland (Germann et al., 2015). The resulting radar products are at high spatial and temporal resolution, with 20 elevation sweeps conducted every five minutes (Germann et al., 2015). The locations and approximate horizontal coverage area of the radar network are plotted in Figures 1 and 2. The reference dataset

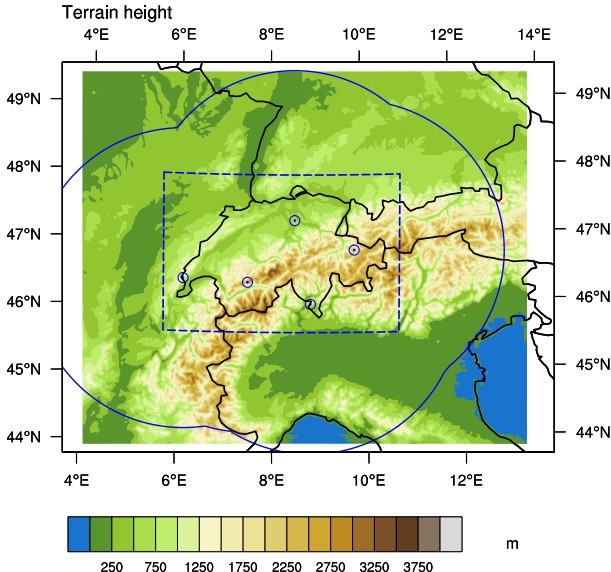

**Figure 2.** As for Figure 1, but for the inner (higher-resolution) nested WRF domain.

we use in this study are results from the TRT algorithm that were compiled into a database of storm cells and their associated properties (as in Nisi et al., 2018, but including data for 2018 and using all Swiss radars).

TRT was developed specifically to deal with the challenging topography of the Alpine region: it takes advantage of the high spatial and temporal resolution of the Swiss radar network (Nisi et al., 2016). TRT identifies thundestorms in a two-dimensional Cartesian multiple-radar "Max Echo" composite product, which is composed of the maximum radar reflectivity recorded in each vertical column (Nisi et al., 2014). TRT uses an adaptive thresholding scheme proposed by Crane (1979) that requires a fixed minimum detection threshold $Z_{min}$ [dBZ], a fixed minimum reflectivity "depth" $Z_{depth}$ [dBZ] and an adaptive threshold

$Z_{thresh}$ [dBZ]. On a two-dimensional map of "Max Echo" radar reflectivity, a cell is defined as a closed contour at $Z_{thresh}$ dBZ, around a maximum reflectivity of $Z_{peak}$ [dBZ]. $Z_{thresh}$ is adapted for each cell to be the minimum value for which $Z_{thresh} \geq Z_{min}$ and for which the cell contains a single closed contour at $Z_{peak} - Z_{depth}$ dBZ (Crane, 1979; Hering et al., 2004). In the case of TRT, $Z_{min}$ is 36 dBZ and $Z_{depth}$ is 6 dBZ, and a further constraint on cell area is applied: for a thunderstorm to be detected by TRT it must contain a connected area of sufficient size with radar reflectivity values at 36 dBZ or higher, and at least one pixel

with a reflectivity of at least 42 dBZ (Hering et al., 2004). The area threshold used in these observations was 13 km$^2$ (Hering, 2020). TRT uses geographical overlapping of cells for matching between time steps (Hering et al., 2004, 2008). Several cell properties are then computed by TRT from the 3D radar data, as well as satellite and lightning data, inside the detected footprint of each cell. A cell severity ranking product is included.

TRT is well tested and established as a reference data set. It has been in operational use at MeteoSwiss since 2003 (Hering

et al., 2008), and formed part of a successful forecast demonstration project in the Alpine region (Rotach et al., 2009). TRT

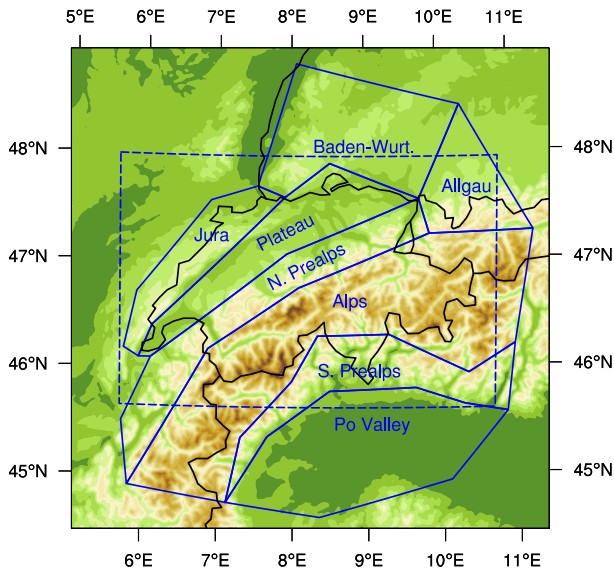

**Figure 3.** Sub-domains used in this study (solid blue lines). Terrain elevation and national borders shown as in Figure 1. "N. Prealps" stands for Northern Prealps, "S. Prealps" stands for Southern Prealps, "Baden-Wurt." stands for Baden-Württemberg. The study domain is shown by the dashed blue line. Plot produced using NCL version 6.6.2.

was used to produce a 15-year, Lagranian-perspective hail climatology for Switzerland (Nisi et al., 2018), as well as to study hailstorm initiation with cold fronts (Schemm et al., 2016). In this study we use TRT results for the study period as the reference data set.

## 2.3 The TITAN storm tracker

TITAN is a radar-based storm cell tracker that uses thresholds on 3D Cartesian fields of radar reflectivity to define contiguous storm areas, for which statistical properties are calculated (Dixon and Wiener, 1993). Matching of storms between time steps is performed using an optimisation algorithm that expects matched storms to have similar volumes and prioritises small separation distance (Dixon and Wiener, 1993). TITAN has been used operationally (e.g. Bally, 2004) as well as in an object-based study of hailstorm properties (Foris et al., 2006).

TITAN (Dixon and Wiener, 1993; TITAN system within LROSE) was downloaded and compiled from the Lidar Radar Open Source Software Environment (LROSE). TITAN uses specialised binary formats for both input and output. As input, TITAN requires data in MDV format with radar reflectivity fields in 3D Cartesian gridded coordinates (Dixon and Wiener, 1993). We used an adapted version of the TITAN tool `NcGeneric2Mdv` to convert input files to MDV format. The output of the tracking process are "storm" files, in which the tracking results are stored in binary format. To extract storm properties from the storm



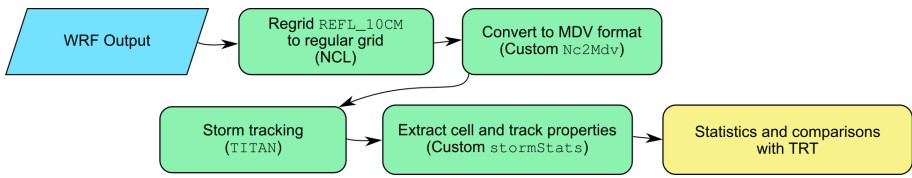

**Figure 4.** The processing flowchart used in this study for WRF data. Shown are input data (blue) processing steps (green), and analyses (yellow). `Nc2Mdv` is a modified version of the TITAN tool `NcGeneric2Mdv`, and `stormStats` is a modified version of the TITAN tool `Storms2Xml2`.

files we used an adapted version of the TITAN `Storms2Xml2` tool. The TITAN processing flowchart for simulation data is shown in Figure 4.

For this study we ran TITAN in dual thresholding mode with auto-restart disabled. In dual-thresholding mode, storms are identified in two steps. First, regions of reflectivity above a lower threshold are identified. Then, within these regions, areas with reflectivities greater than a sub-region reflectivity threshold are identified, tested for size, and "grown" out into the original lower-threshold region (Dixon and Seed, 2014). Threshold choice is discussed in Section 2.6.

## 2.4 WRF weather model

WRF is a weather model used for both research and operational NWP (Skamarock et al., 2019; Powers et al., 2017). When run at sufficiently high spatial resolution, it can explicitly resolve convection. What constitutes a sufficient resolution depends on the application: model grid spacings finer than 1 km are optimal for resolving all convective processes, while proper resolution of turbulent processes requires a grid spacing in the order of 100 m (e.g. Bryan et al., 2003; Bryan and Morrison, 2012). However, grid spacings up to 4 km provide enough detail to explicitly resolve basic cumulous cloud structures (e.g. Weisman et al., 1997; Done et al., 2004; Kain et al., 2006; Chevuturi et al., 2015). In this work we ran WRF with 50 vertical levels, on a regional rotated grid, with average horizontal resolution of about $1.5 \times 1.5$ km$^2$. A nested domain structure was used with a larger external domain at an average of about $4.6 \times 4.6$ km$^2$ resolution. The two domains are shown in Figures 1 and 2 respectively.

We used WRF version 4.0.1 (Wang et al., 2018). HAILCAST (Brimelow et al., 2002; Adams-Selin and Ziegler, 2016) was used to calculate maximum hail sizes. We tested three different WRF microphysics schemes: the Predicted Particle Property (P3) scheme (Morrison and Milbrandt, 2015), the Morrison scheme (Morrison et al., 2009) and the Thompson scheme (Thompson et al., 2008). The other physics schemes used in the model are shown in Table 2. The boundary data used were European Centre for Medium-Range Weather Forecasts (ECMWF) operational analyses from the Integrated Forecasting System (IFS) cycle 43r3 (ECMWF, 2017; Buizza et al., 2017). Radar reflectivity was calculated by WRF, with the option `do_radar_ref` enabled to instruct WRF to calculate reflectivity using microphysics-scheme specific parameters (Wang et al., 2018). The simulations covered a time period from May 1, 2018 to May 31, 2018 at five-minute resolution.





**Table 2.** Schemes used in the WRF model in this study.

| Configuration option | Scheme used |
| --- | --- |
| Boundary layer scheme | Yonsei University (Hong et al., 2006) |
| Cumulus parameterisation | None (explicit convection) |
| Shortwave radiation scheme | Dudhia (Dudhia, 1989) |
| Longwave radiation scheme | RRTM (Mlawer et al., 1997) |
| Land surface scheme | Noah (Chen and Dudhia, 2001) |
| Surface layer model | Revised MM5 Monin-Obukhov (Jiménez et al., 2012) |
| Hail model | HAILCAST (Adams-Selin and Ziegler, 2016) |

Storm-tracking was run on the WRF output variable `REFL_10CM`, which contained estimated 10 cm wavelength (S-band)

radar reflectivity in dBZ. The WRF data were treated using an NCAR command language (NCL, version 6.4.0, NCL6.4) script to regrid the data to a Cartesian grid stored in NetCDF format. The files were first regridded horizontally by dividing the WRF domain into a grid with the same number of points and extents of latitude/longitude values as the input fields, but with the points as evenly spaced as possible on each axis. The regridding was performed using bilinear interpolation provided by the Earth System Modeling Framework (ESMF, version 8.0.0, Valcke et al., 2012) through NCL. The output grid had a

resolution of approximately 0.0141° latitude by 0.0211° longitude. This grid was then interpolated vertically using the NCL `wrf_user_vert_interp` function to grid points from 1 km to 15 km above sea level at 0.5 km resolution. These heights were geopotential heights above sea level; the small differences between geopotential and geometric heights are ignored in this study. Interpolation of radar reflectivities was performed using dBZ values. The regridded WRF files were converted to MDV format for use with TITAN.

**2.5 Comparing storm properties**

Before comparisons of tracking results were made, TRT and TITAN cell detections with centre points outside the study domain (see Figure 2) were discarded. Cells that were truncated by this operation had their durations shortened to the duration for which they stayed within the region of interest. Likewise, cells that were split into multiple parts by the spatial subsetting operation were updated so that their parts were counted as separate storm cells.

Thunderstorms often split into multiple parts or merge from multiple parts into single cells. TITAN and TRT handle the labelling of these storms differently. TITAN data contain a "storm ID" that is maintained through splits and merges, and a "track ID" which refers to a unique length of storm track with no splits or merges. TRT data contain flags indicating when splits and merges have occurred, and the most intense storm part keeps the same identifier afterwards. Due to these labelling differences, in this paper we take a simplified approach and refer to a "cell" as a region of high radar reflectivity that exists for

at least 30 minutes with no splitting or merging events. When a split occurs, the parent cell ends and multiple new (child) cells are created, and when a merge occurs multiple cells end and a new (merged) cell is created. In this way we lose information on





the overall length of one storm system, but we can compare cell properties easily and fairly. A "track" is the path over which a cell moves. A "cell detection" refers to a region of high reflectivity at one moment in time. Some storm properties (area, movement direction) are defined for each cell detection, while some (duration) are defined for each cell.

The TRT results are taken as the reference data set, and TITAN results were compared to the TRT database to analyse the performance of the TITAN approach. The comparison measures used were defined as follows: for a given storm property $P$, let $P_{i,\text{TITAN}}$ be the $i$th value of the property given by the TITAN approach and let $P_{i,\text{TRT}}$ be the corresponding $i$th reference value of the property in the TRT database ($i$ refers to an index shared by both datasets, such as simulation day). The difference between the two results is given by

$$D_i = P_{i,\text{TITAN}} - P_{i,\text{TRT}}. \tag{1}$$

The bias of the TITAN approach is $\langle D \rangle$, where the angular brackets signify the mean of all differences. The root mean squared error (RMSE) is $\sqrt{\langle D^2 \rangle}$. The relative error is given as a percentage by

$$R_i = \frac{100 D_i}{P_{i,\text{TRT}}}. \tag{2}$$

The mean relative bias (RB, $\langle R \rangle$), the median relative bias (MRE, median of $R$), and the interquartile range of relative bias (RE IQR, 75th percentile minus 25th percentile of $R$) to measure relative differences. The Pearson correlation coefficient ($r^2$) is used to show the cofluctuation of $P_{\text{TITAN}}$ and $P_{\text{TRT}}$. The relative error is only defined when $P_{i,\text{TRT}}$ is non-zero; accordingly RB, MRE and RE IQR include only data points for which $P_{i,\text{TRT}} \neq 0$, whereas bias, RMSE, and $r^2$ include such points. Days on which no technique identified cells are not counted in the statistics.

### 2.6 Optimisation of TITAN thresholds

Radar reflectivities simulated in WRF at S-band are not expected to match the measured radar reflectivities at C-band that were used by TRT, so we did not attempt to make TITAN use exactly the same thresholds as TRT. Furthermore, the TRT detection works on two-dimensional fields and thresholds on cell area, whereas TITAN uses three dimensional fields and thresholds on cell volume. Our simulation setups differed only in the microphysics scheme used, but since the calculation of radar reflectivities can be affected by the microphysics scheme, optimum thresholds were expected to differ between simulation sets.

We chose to optimise three TITAN thresholds by finding the values that provided the best match between TITAN+WRF (simulation) output and TRT results (observations) for May 29 and 30, 2018, two days over which thousands of storm detections were made across the domain. The optimised thresholds were then used for validation of the technique on the whole dataset of May 2018. The three thresholds tested were: 1) the reflectivity threshold for cell detection, (`low_dbz_threshold` in the TITAN parameter file), with tested values from 34 dBZ to 42 dBZ in 1 dBZ increments; 2) the reflectivity threshold for dual-thresholding (`dbz_threshold` under `dual_threshold`), with tested values from `low_dbz_threshold` plus 4 to



`low_dbz_threshold` plus 12 dBZ in 1 dBZ increments; and 3) the volume threshold for cell detection `min_storm_size`, with tested values of 25, 50, and 75 km$^3$.

TITAN was run on WRF output for the test days with all 243 tested combinations of the three thresholds. The results for
each run were compared to TRT results for those days. The "best" parameter set was non-trivial to select and depended on the performance statistics used. We chose an approach that emphasised low bias and cofluctuation in simulated and observed number of cells per hour, and a good match on cell area. To choose the "winning" parameter set we used absolute value of median relative bias as a score. This score was applied to comparisons of daily median cell area, and per time-step number of cells. We first subset based on number of cells per hour by taking all test runs with scores less than the 10th percentile
of all scores. We then subset based on daily median cell area by again taking scores less than the 10th percentile of all such scores. Of the few remaining tested combinations we chose the configuration with the best squared correlation coefficient value for simulated and observed per time-step number of cells. The resulting thresholds used for TITAN tracking in this study are shown in Table 3. Reports showing details of the threshold testing are archived (Raupach et al., 2021d).

Other parameters in the dual-thresholding scheme were held fixed for all model runs. These parameters were the mini-
mum area required for each sub-part in the dual thresholding approach (`min_area_each_part`), which was set to 16 km$^2$, the fraction of the lower-reflectivity storm region that must be covered by the sum of all higher-reflectivity sub-regions (`min_fraction_all_parts`), set to 0.10, and the minimum proportion of the large area that each sub-area must exceed (`min_fraction_each_part`), set to 0.005. These last two area thresholds are those listed in the default TITAN parameters as appropriate for strong convection and squall lines in South Africa[1].

## 3   Results

In this section, storm properties found using TITAN on WRF simulation output are compared to those found using TRT on radar data, to test whether TITAN applied to WRF simulations can produce representative statistics on thunderstorms in Switzerland. TITAN was run over the WRF simulation outputs, and TRT results were subset to the same period of time. Both sets of results were subset to the study domain shown by the dashed line in Figures 1 and 2. During subsetting of the
TITAN (TRT) results, including all tested microphysics scheme setups, subsetting caused splits in 0.78% (0.64%) of cells. After subsetting, 37.8% (52.4%) of the recorded cells were discarded because their track duration was less than 30 minutes. The resulting cell descriptions from TITAN sometimes contained spatial overlaps; 23% of cells were affected by overlaps, but the areas affected were small with only 3% of all cell points overlapping. Of the TRT cells remaining after subsetting, 30 (0.06%) were removed from this analysis because no cell velocity information was recorded.

Table 4 shows a comparison of the number of detections (here defined as a unique storm/time combinations) and storm cells captured by each technique. When each microphysics scheme was compared to the reference TRT dataset, TITAN produced 15% more detections for the Morrison scheme, 9% fewer detections for the P3 scheme and 15% fewer detections for the

---

[1]Stated in the TITAN `paramdef.TITAN` file, https://github.com/NCAR/lrose-core/blob/master/codebase/apps/titan/src/Titan/paramdef.Titan, accessed 23.12.2019.





**Table 3.** The threshold values used in each application of TITAN. Other thresholds were left at default values. These thresholds are for the basic detection threshold ($Z$ threshold, the `low_dbz_threshold` parameter), the dual-thresholding sub-region threshold (Sub-region $Z$ threshold, `dual_threshold`'s `dbz_threshold` parameter), the minimum allowed storm volume (Min. volume, the `min_storm_size` parameter), and the minimum area for sub-parts (Min. sub-area, `dual_threshold`'s `min_area_each_part` parameter).

|  | $Z$ threshold [dBZ] | Min. sub-area [km$^2$] | Min. volume [km$^3$] | Sub-region $Z$ threshold [dBZ] |
|---|---|---|---|---|
| Morrison | 42 | 16 | 50 | 54 |
| P3 | 39 | 16 | 50 | 47 |
| Thompson | 40 | 16 | 75 | 47 |

**Table 4.** Summary information for each data set, showing the number of cell detections (cell/time combinations), number of cells, and first and last cell detection times.

| Method | Num. detections | Num. cells | First cell (UTC) | Last cell (UTC) |
|---|---|---|---|---|
| WRF+TITAN (Morrison) | 29865 | 2708 | 2018-05-01 00:00 | 2018-05-31 23:55 |
| WRF+TITAN (P3) | 23696 | 2292 | 2018-05-02 22:35 | 2018-05-31 23:55 |
| WRF+TITAN (Thompson) | 21974 | 2301 | 2018-05-03 04:40 | 2018-05-31 23:35 |
| TRT (Observations) | 25921 | 2831 | 2018-05-02 19:10 | 2018-05-31 23:55 |

Thompson scheme. TITAN produced 4% fewer cells for the Morrison scheme, 19% fewer cells for the P3 scheme and 19% fewer cells for the Thompson scheme than were in the TRT dataset. In the rest of this section, we show detailed comparisons
with sub-regions identified as shown in Figure 3. The thunderstorm properties are divided into four categories: spatial and temporal cell occurrences (Section 3.1), cell movement properties (Section 3.2), hail properties (Section 3.3), and storm lifecycle properties (Section 3.4).

### 3.1 Spatial and temporal cell occurrences

Figure 5 shows a comparison of the number of storm detections (cell/time combinations) per $10 \times 10$ km$^2$ raster grid point, to
show the "hot spots" of storm activity during the month of May 2018 in both the simulations and observations. The figure shows broadly similar spatial layouts between observations and simulations. In particular, the observations and all simulations show regions of increased storm occurrence over the northern flanks of the Jura mountains that run along the border of Switzerland and France; southeastern Germany; the southern Swiss Plateau and northern prealps; and northern Italy to the east of Ticino (the part of Switzerland that extends into the "Southern Prealps" region shown in figure 3). The simulated storm hot spots
over the Jura are to the north of the observed Jura hotspot. Notably, the simulations all underestimate the concentration of storm detections in Ticino observed by radar. The simulations all reproduce the minima of storm activity that traces the main

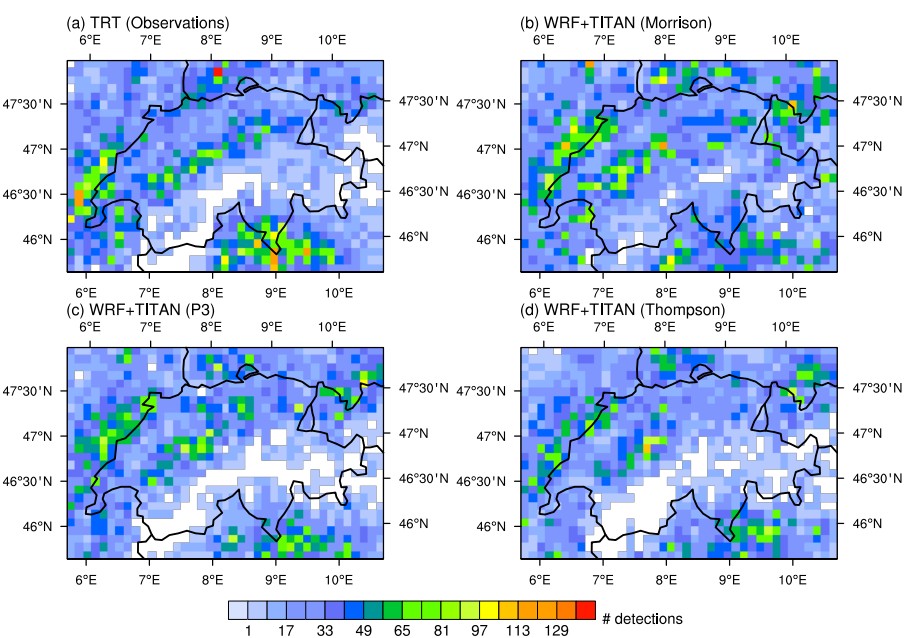

**Figure 5.** The overall number of cell detections (cell/time combinations) in each $10 \times 10$ km$^2$ grid point, for May 2018, for observations (a) and simulations with three different microphysics schemes (b-d). Plot produced using NCL version 6.6.2.

Alpine range; in this regard, the P3 and Thompson schemes produce more realistic maps than the Morrison scheme. Overall, the approach of using TITAN on WRF output is able to broadly reproduce the observed locations of cell detection maxima.

Figure 6 shows the number of cells detected by each technique on each day of May 2018. Table 5 shows statistics to compare of number of cells per day between the simulations and observations. Because the simulations and observations are independent and the simulations are forced only by lower-resolution boundary conditions, we do not necessarily expect an exact match in cell occurrence timeseries. The simulated number of cells detected per day show similar magnitudes to the observations, with exceptions in Allgau, the Alps for the Morrison scheme, and the Po valley for the P3 scheme, where more cells were detected in the simulations. In terms of median relative bias, the best per-region performance was with the Thompson scheme in the Alps region (-2%), and the best performance for all regions combined was with the Morrison scheme (-13%). The worst overall match was with the P3 scheme (-20%). The worst per-region median relative bias was with the Morrison scheme in the Alps region (78%). The greatest co-fluctuation ($r^2$ value) in a single region was shown by the Thompson setup in the Alps region (0.74), and overall by the Thompson scheme (0.56). That positive correlations exist on cells per day shows that the WRF model is able to use these boundary conditions to produce thunderstorm cells on storm-prone days.

To investigate any systematic timing differences and to look at the diurnal cycle of the thunderstorms, we calculated the percentage of cells that appeared in each hour of the day, for each simulation and for the observations. These results are



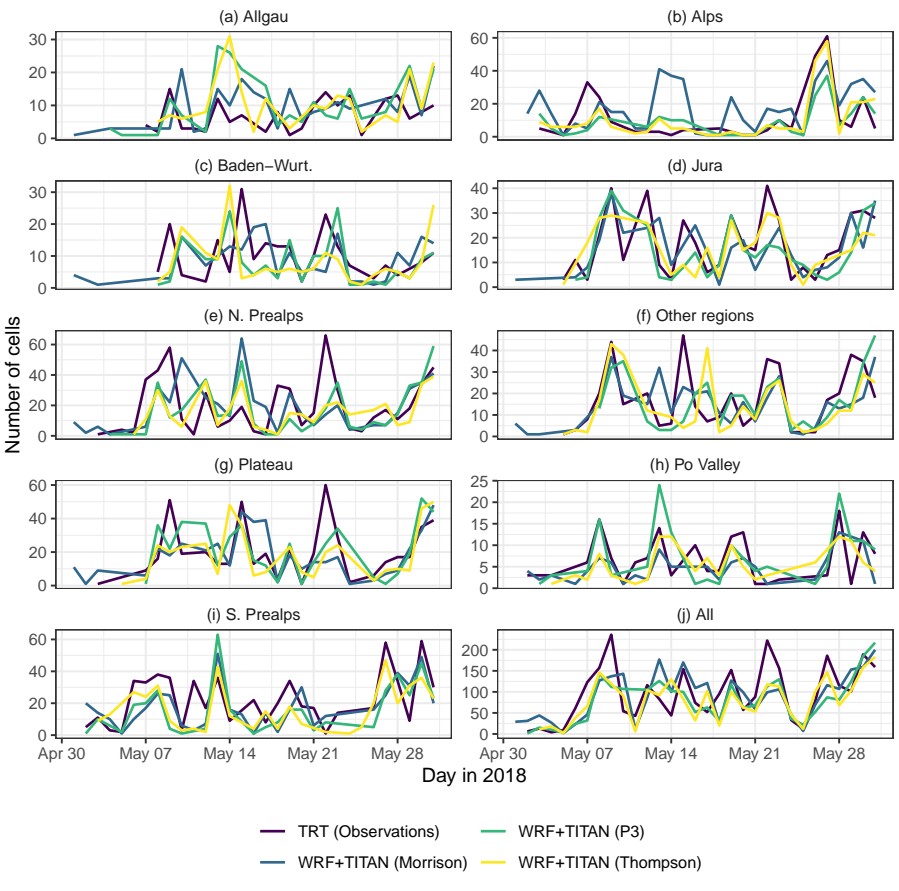

**Figure 6.** The number of cells detected per day in May 2018, for observations and simulated outputs, per region (regions shown in Figure 3.

shown by region in Figure 7. In all regions, the afternoon peak in thunderstorm activity is well reproduced by the simulations, although the exact timings differ from the obsevations in some regions. There is a tendancy for the Morrison and P3 simulations to produce more cells during the night time than are observed, and this continues into the morning for the Morrison scheme.

For all data, the peak time for cell occurrence in the Thompson simulations matches the peak time in the observations, while the peak in the Morrison set is one hour earlier and there are peaks in the P3 scheme both one hour earlier and one hour later than the observed peak at 15 UTC. There is an interesting pattern in the results in which simulated storms tend to appear earlier than the observed storms in the north and northwest (Jura, Allgau, Other regions), at about the same time as the observations in central Switzerland (Alps, N. Prealps, Plateau), and later than the observations in the southern Prealps. The results for the

Po Valley match well to observations. Earlier storms in the north and later storms in the south have been shown in previous radar-based climatologies (Nisi et al., 2016), but here this effect is more extreme in the simulations than in the observations. The north to south differences are possibly due to different handling of convective initiation mechanisms in the weather model.



**Table 5.** Performance statistics on cells detected per day per region, with TRT (Observations) taken as the reference. Statistics shown are bias [d$^{-1}$], root mean squared error (RMSE) [d$^{-1}$], relative bias (RB) [%], median relative error (MRE) [%], interquartile range of relative error (RE IQR) [% points], and squared Pearson correlation ($r^2$) [-].

|  |  | Bias | RMSE | RB | MRE | RB IQR | $r^2$ |
|---|---|---|---|---|---|---|---|
| WRF+TITAN (Morrison) | Allgau | 2.8 | 7.7 | 126 | 10 | 142 | 0.04 |
|  | Alps | 8.1 | 16.8 | 394 | 78 | 448 | 0.20 |
|  | Baden-Wurt. | −0.3 | 8.4 | 34 | −15 | 178 | 0.07 |
|  | Jura | −1.5 | 9.5 | 26 | −10 | 132 | 0.45 |
|  | N. Prealps | −1.0 | 19.1 | 110 | −15 | 145 | 0.13 |
|  | Other regions | −1.4 | 11.3 | 28 | −16 | 91 | 0.38 |
|  | Plateau | −0.8 | 14.1 | 70 | −10 | 81 | 0.30 |
|  | Po Valley | −1.7 | 4.7 | 36 | −33 | 50 | 0.28 |
|  | S. Prealps | −4.4 | 15.1 | 51 | −31 | 129 | 0.28 |
|  | All | −3.7 | 52.5 | 44 | −13 | 109 | 0.41 |
| WRF+TITAN (P3) | Allgau | 3.0 | 8.6 | 115 | 12 | 212 | 0.11 |
|  | Alps | −2.5 | 10.5 | 47 | −15 | 190 | 0.57 |
|  | Baden-Wurt. | −1.8 | 9.4 | 21 | −33 | 90 | 0.03 |
|  | Jura | −3.3 | 9.6 | −4 | −16 | 64 | 0.50 |
|  | N. Prealps | −4.1 | 17.6 | −7 | −9 | 106 | 0.24 |
|  | Other regions | −2.1 | 13.6 | 5 | −27 | 93 | 0.23 |
|  | Plateau | −0.9 | 13.5 | −7 | −12 | 73 | 0.40 |
|  | Po Valley | −0.2 | 4.8 | 49 | −29 | 97 | 0.49 |
|  | S. Prealps | −7.9 | 15.6 | −7 | −44 | 68 | 0.39 |
|  | All | −18.3 | 49.2 | −6 | −20 | 71 | 0.52 |
| WRF+TITAN (Thompson) | Allgau | 2.3 | 8.3 | 81 | 30 | 188 | 0.09 |
|  | Alps | −0.3 | 7.7 | 37 | −2 | 75 | 0.74 |
|  | Baden-Wurt. | −1.3 | 10.9 | 48 | −40 | 90 | 0.00 |
|  | Jura | −3.2 | 9.8 | 18 | −26 | 94 | 0.45 |
|  | N. Prealps | −5.8 | 16.0 | −4 | −33 | 102 | 0.31 |
|  | Other regions | −3.3 | 14.3 | 16 | −20 | 104 | 0.20 |
|  | Plateau | −3.2 | 14.1 | 8 | −44 | 81 | 0.37 |
|  | Po Valley | −1.5 | 5.0 | 29 | −50 | 52 | 0.21 |
|  | S. Prealps | −6.6 | 13.3 | −11 | −27 | 90 | 0.51 |
|  | All | −18.2 | 46.8 | −1 | −19 | 57 | 0.56 |



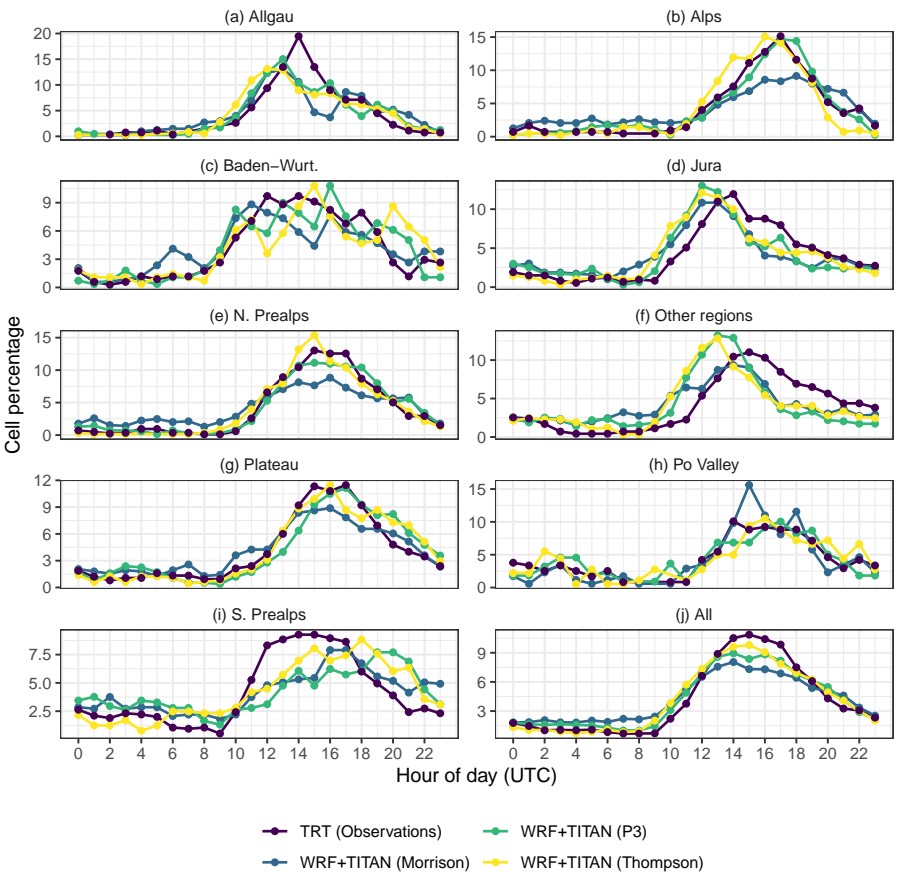

**Figure 7.** Percentages of cells that were active in each hour of the day, per region, in May 2018, with observations compared to simulation outputs. Values are the percentage of all unique cell/hour combinations that occurred in each individual hour of the day, so that values for each curve sum to 100.

There are known differences in storm initiation between northern regions of Switzerland and regions on the south of the main Alpine chain (Nisi et al., 2016, and references within).

## 3.2 Cell movement properties

The use of object-based analysis means we can compare aggregate storm properties such as movement speed, direction, intensity, or cell lifetime. Figure 8 shows a comparison of the directions in which detected cells were moving at each observation point. Although there are some differences in the proportions between TRT and TITAN, it is notable that the simulations are able to reproduce the differences in advection direction observed between different regions. For example, the TRT observations show that storms moved mostly in a north and northwest direction in the Po Valley, and in a southwest direction on the Swiss plateau. The simulations reproduce these differences. Again, the region of Allgau shows notable differences between observa-





**Table 6.** Mean advection directions by region.

| | Mean angle (degrees) | | | |
|---|---|---|---|---|
| Region | TRT (Observations) | WRF+TITAN (Morrison) | WRF+TITAN (P3) | WRF+TITAN (Thompson) |
| Allgau | 314 | 17 | 319 | 323 |
| Alps | 349 | 358 | 334 | 359 |
| Baden-Wurt. | 246 | 310 | 287 | 312 |
| Jura | 322 | 344 | 350 | 357 |
| N. Prealps | 301 | 24 | 348 | 352 |
| Other regions | 269 | 292 | 309 | 284 |
| Plateau | 261 | 316 | 290 | 306 |
| Po Valley | 322 | 305 | 327 | 348 |
| S. Prealps | 333 | 340 | 344 | 338 |
| All | 295 | 340 | 327 | 332 |

tions and simulations. Table 6 shows the mean direction of all cells by region and dataset. The simulation set that produced the best match with observations differed by region, but the P3 scheme produced the best match in more regions than the other simulation sets.

Figure 9 shows quantile-to-quantile (QQ) comparisons of three other properties: cell detection areas, cell detection velocities, and cell durations. We consider very high velocities ($> 80$ km h$^{-1}$) to be unrealistic artefacts of the tracking algorithms; for both TRT and TITAN+WRF results less than 0.5% of cell detections had such velocities. We note again that these durations are the durations of cells as defined here, meaning that they are interrupted by storm splits and merges. The QQ plots map observed quantiles of these properties to simulated quantiles, over all detected cells. If the simulated distributions match the

observed distributions, the lines follow the diagonal (solid black) line on the QQ plot. The plot shows that the simulated distributions broadly agree with observed distributions for velocity in all simulations and for area and duration for the P3 and Thompson microphysics scheme setups. For the simulations run with Morrison microphysics, the plot shows that the detected cell areas were larger than the observed cells, and the simulated cells lasted for longer durations than the observed cells. Cell area and duration is most affected by the choice of thresholds used in the TITAN tracker, which means that these differences

are unlikely to be caused by the microphysics scheme as such, but rather by the thresholds that result from the optimisation process described in Section 2.6.

### 3.3   Hail properties

In this section we compare radar-based observations of hail properties to those estimated by the WRF model and HAILCAST. The object-based technique we test here may be particularly useful for studying the effects of climate change on hail, on which

there remains high uncertainty (e.g. Raupach et al., 2021b). In each dataset we compare the proportion of storm cell pixels that





**Table 7.** Proportions of total cell detections that contained hail with estimated diameter greater than 25 mm or 40 mm, for observations and simulation outputs.

|  | Proportion of cell detections with hail over | |
| --- | --- | --- |
|  | 25 mm [%] | 40 mm [%] |
| TRT (Observations) | 3.2 | 1.3 |
| WRF+TITAN (Morrison) | 1.5 | 0.2 |
| WRF+TITAN (P3) | 1.6 | 0.5 |
| WRF+TITAN (Thompson) | 2.5 | 0.4 |

were estimated to contain severe (greater than 2.5 cm) and very large (greater than 4 cm) hail. In the observations from TRT, the maximum hail size was estimated using the radar-based maximum expected severe hail size (MESHS, implementation described in Nisi et al., 2016). In the WRF output, we used the HAILCAST variable `HAILCAST_DIAM_MAX` to calculate the proportions of TITAN-identified cell pixels with hail over 2.5 cm and 4 cm respectively. We note that the two techniques used
to estimate maximum hail size are very different from each other and are therefore not strictly directly comparable; they are used here as the available approximations of observed and simulated hail size.

Table 7 shows the proportions of all cell detections that contained severe hail. In general, the observations contained more severe hail than the simulations. All WRF setups underestimated the proportion of cell detections containing severe hail. The WRF setup using the Thompson microphysics scheme produced the closest match to the TRT proportion of cell detections
with hail over 2.5 cm. The relative errors on these proportions were smaller for 2.5 cm hail than for 4 cm hail, implying that the WRF and HAILCAST simulations more severely underestimated the number of cells containing very large hail than severe hail. Figures 10 and 11 show quantile-to-quantile plots to compare the proportions of cell pixels, for cell detections for which the proportion was non-zero, that contained hail with maximum estimated size over 2.5 cm and over 4 cm respectively. The WRF results show an understimation of the cell area covered by severe and large hail, compared to the TRT observations.

**3.4   Cell lifecycles**

In this section we consider cell lifecycles – the evolution of the strength of storm cells over their durations. Since in this work splits and merges of storms interrupt storm durations, in this section we consider only the 43% of cells that contained no splits or merges, so that their durations are well defined. Figure 12 shows the number of such cells by cell duration. There are very few cells with duration over 100 minutes, meaning little emphasis should be placed on aggregate results for these long-duration
cells. Figure 13 shows the development of cell area over time. The WRF simulations match the TRT observations well, with the exception of the Morrison scheme setup for which areas are overestimated at all points in the cell's lifecycle. We emphasise, though, that since the area of cells at detection is defined by a threshold on storm size, the difference here has more to do with our optimised TITAN threshold values than with the microphysics scheme itself. The Thompson and P3 scheme setups provide a close match for cells up to about 100 minutes from their starting time. In Figure 14, relative intensities of cells are compared





to the relative positions in the cells' durations. Cells tracked in the simulations achieve their maximum intensities earlier than the observed cells, but decay in a similar way. Differences between the different WRF setups are primarily in the first and last thirds of the storm lifecycle, with the P3 scheme setup showing higher earlier intensities and earlier decay, and the Morrison results showing the best match with observations from halfway through the track durations to about 85% through the durations.

## 4 Conclusions

In this study we tested and verified an approach for the object-based analysis of simulated thunderstorms in the topographically complex Alpine region of Europe. Output from a high-resolution weather model (AR-WRF) was analysed using a radar storm tracking system (TITAN) to derive characteristics on each storm cell. The results were compared to a reliable and independently derived dataset of storm observations for Switzerland (TRT) for the month of May 2018. We tested WRF and TITAN using three different microphysics schemes.

The choice of radar reflectivity and cell volume thresholds to use in TITAN made a significant difference to the quality of the results. We optimised the thresholds to find the best settings to use for each microphysics scheme, but this search was location-dependent and not exhaustive, the resulting thresholds depended on which performance criteria were emphasised, and the search-space over which thresholds are optimised could be further refined. The results of this study should thus not be seen as a comparison of the physical appropriateness of the microphysics schemes, but a comparison of three possible setups (comprising

both scheme and chosen thresholds) for summarising thunderstorm properties in simulations over the Alpine region. TITAN thresholds, including those not optimised here such as the dual-thresholding scheme settings, should be carefully considered in any work that uses this technique. We used a simplified approach in which splits and merges in storm cells were ignored. Future work could take splits and merges into account in order to properly characterise full storm lifecycles. Updates to TITAN have been suggested (e.g. Han et al., 2009; Muñoz et al., 2018), and could also be tested in future studies. We showed comparisons

for simulated and radar-derived hail properties; in future, liquid precipitation could also be considered through the use of disaggregated precipitation fields (e.g. Barton et al., 2020).

The goal of this study was to determine whether TITAN plus WRF can provide a realistic representation of thunderstorm activity in Switzerland. The results show a reasonable match between simulated and observed storm properties can be obtained, if thresholds for TITAN cell detection are carefully chosen. The level of agreement between TITAN and WRF, for geographic

distribution, diurnal cycle, number of cells per day, hail properties, and cell area, duration, velocity, and movement direction, shows that WRF is able to explicitly resolve thunderstorm cell properties to an acceptable standard of accuracy at $\sim$1.5 km$^2$ resolution, over a topographically complex region. Further, the approach of using TITAN to analyse storm properties produces results that are representative enough of the current climate to justify continuing use of the technique for comparisons between simulations of current and future scenarios. This technique therefore holds promise for investigation of how convective storms,

including hailstorms, may be affected by climate change.



*Code and data availability.* Code for this project is available under the MIT license at https://github.com/traupach/stormtrack. Modified versions of LROSE utilities are available under the LROSE BSD license at https://github.com/traupach/modified_LROSE_utils. Any code updates will be posted at these GitHub addresses. The exact versions of the code used to produce the results shown here are available as Zenodo archives for the original code (Raupach et al., 2021d, MIT license), modified LROSE tools (Raupach et al., 2021c, LROSE BSD

license), and R Markdown for this manuscript (Raupach et al., 2021e, CC-BY-4.0 license). TITAN tracking data and hail statistics extracted from WRF outputs are archived on Zenodo (Raupach et al., 2021a, CC-BY-4.0 license). Fields of extracted WRF output (simulated radar reflectivity and maximum HAILCAST hail size) are archived on Zenodo for the Morrison (Martinov et al., 2021a, CC-BY-4.0 license), P3 (Martinov et al., 2021b, CC-BY-4.0 license) and Thompson (Martinov et al., 2021c, CC-BY-4.0 license) microphysics schemes. Other WRF model output data are available from the authors by request. TRT data are proprietary to MeteoSwiss and are not publically available; the

contact details for MeteoSwiss are listed online (MeteoSwiss, 2021).

*Author contributions.* TR and OM designed the study. TR performed the analyses and wrote the manuscript. AM configured and ran WRF to produce model output. LN and AH provided expert advice on TRT. YB compiled TRT data. All authors provided feedback on the manuscript.

*Competing interests.* TR (until 31.12.2019), OM (ongoing), AM (ongoing), and YB (ongoing) were in positions funded by the Mobiliar Insurance Group. This funding source played no role in any part of the study.

*Acknowledgements.* The authors thank MeteoSwiss and Urs Germann for providing the TRT storm database. Simulations were calculated on UBELIX (http://www.id.unibe.ch/hpc), the HPC cluster at the University of Bern.





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



**Figure 8.** Comparison of tracked cell directions by TITAN (on WRF data) and TRT (on radar observations). Shown are the percentage of times cells were detected as moving in each of eight compass directions, by dataset.





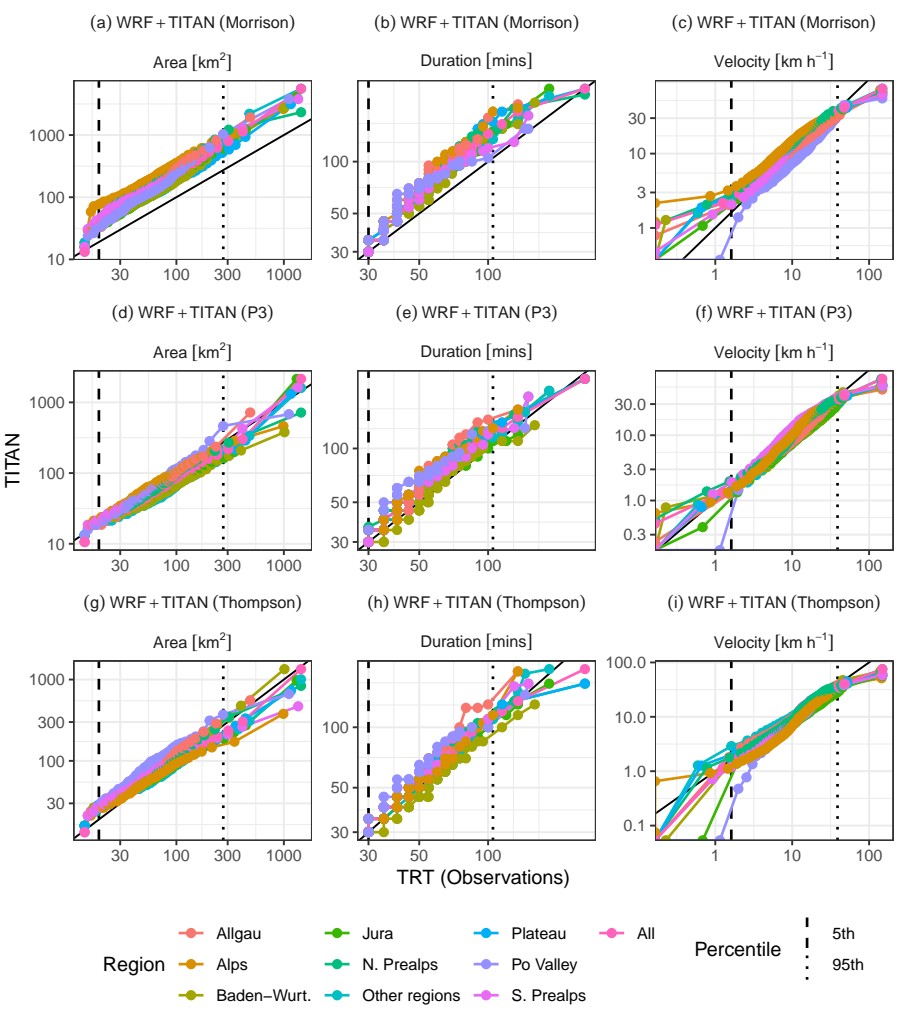

**Figure 9.** Quantile-to-quantile (QQ) comparisons of cell detection areas, cell detection velocities, and cell durations by TITAN (on WRF simulations) and TRT (on radar observations). The black solid line is the 1:1 line. The vertical dashed lines show the 5th and 95th percentiles in the TRT distributions. Since the distribtions are skewed, these plots are on logarithmic axes (zeros are plotted on the axes); the same plot with linear axes is shown for comparison in Figure A1.





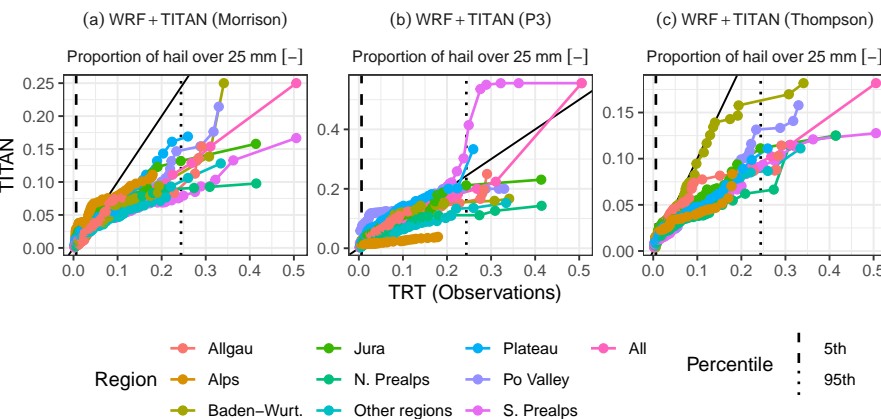

**Figure 10.** Quantile-to-quantile (QQ) comparisons of the proportion of pixels with maximum hail size over 25 mm, for cell detections for which this proportion was greater than zero. The black solid line is the 1:1 line. The vertical dashed lines show the 95th and 99th percentiles in the TRT distributions.

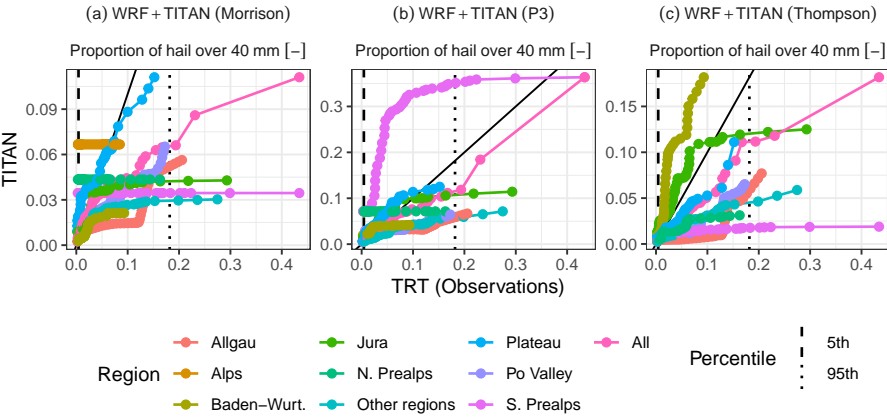

**Figure 11.** As for Figure 10 but for maximum hail size over 40 mm.





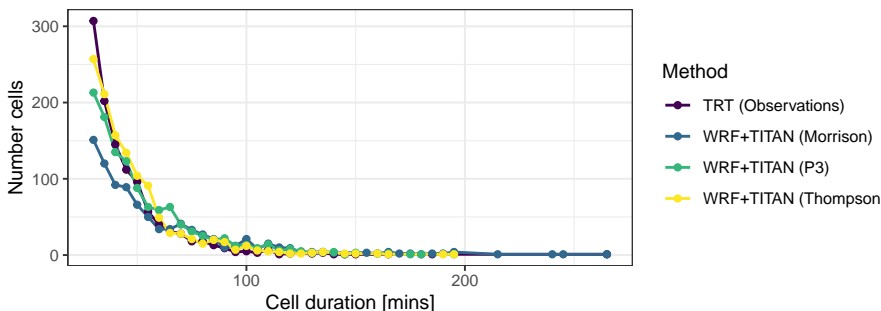

**Figure 12.** The number of cells detected, by cell duration, for observations and simulation outputs.

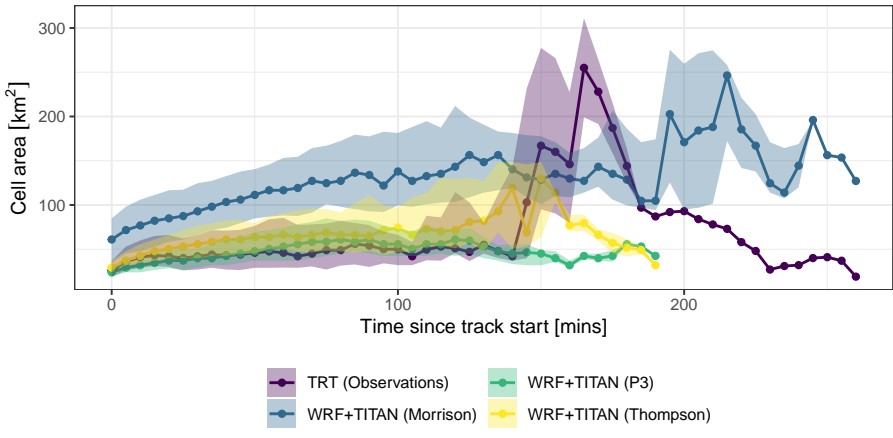

**Figure 13.** Area development over cell lifecycles for observations and simulation outputs. For each time since the track start, the coloured band shows the inter-quartile range of area and the joined points show the median area.

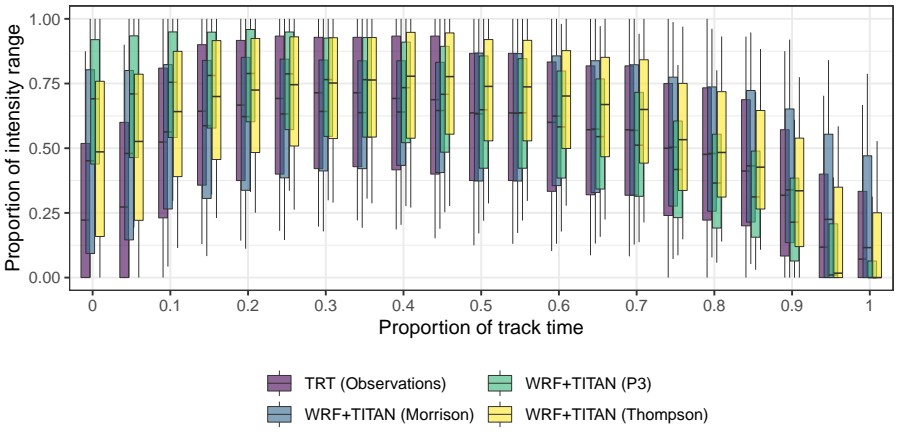

**Figure 14.** Relative lifecycle of storm cells.

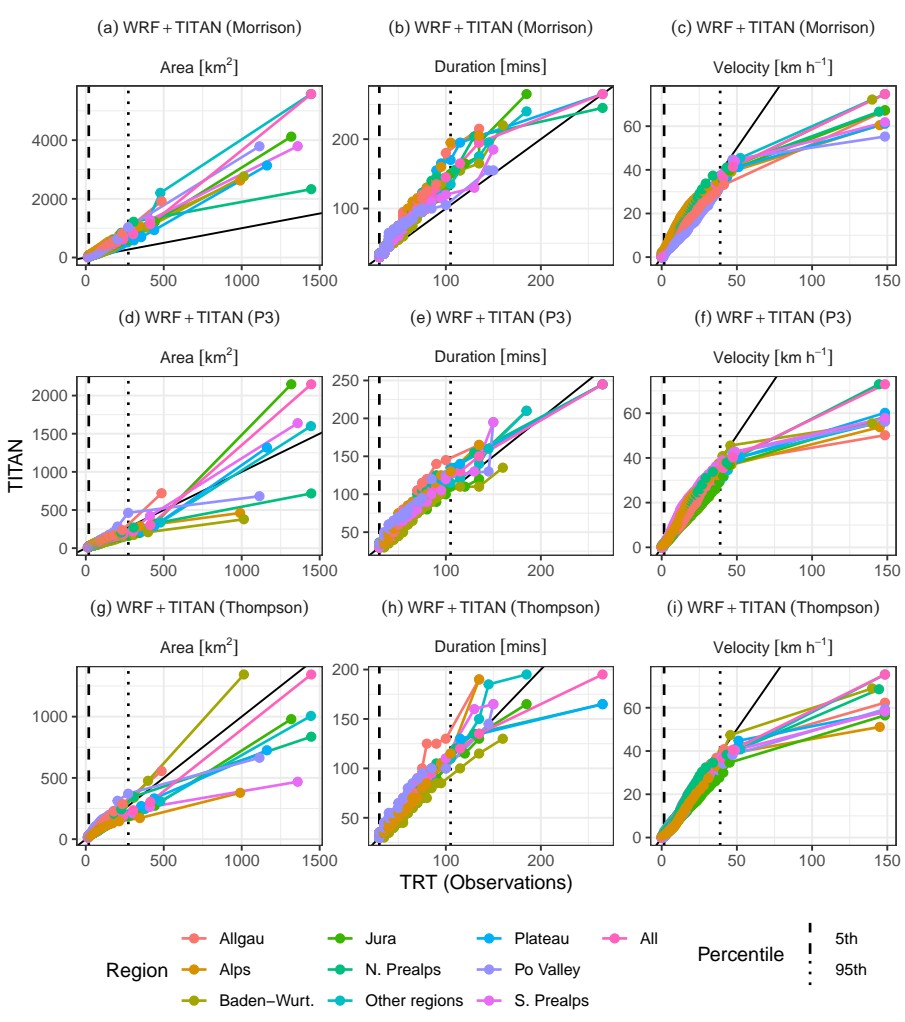

**Figure A1.** As for Figure 9, but with quantiles plotted on linear scales.