# Peer review of "Object-based analysis of simulated thunderstorms in Switzerland: application and validation of automated thunderstorm tracking on simulation data"

_Geoscientific Model Development, 2021_

## Author Comment (AC1)

**gmd-2021-105: responses to reviewers**

Timothy H. Raupach, Andrey Martynov, Luca Nisi,
Alessandro Hering, Yannick Barton, and Olivia Martius

August 10, 2021

In this document we provide responses to reviewers of our manuscript titled "Object-based analysis of simulated thunderstorms in Switzerland: application and validation of automated thunderstorm tracking on simulation data". The reviewers' comments are reproduced in italics, verbatim with the exception that the reference format has been homogenised and references are listed at the end of this document.

**Reviewer 1**

*This study uses a newly developed tracking method to evaluate the WRF model simulation of storm over Swiss region. This is a quite important topic as when the model resolution goes higher and higher, finally, storms can be direcly resolved in the models but how to evaluate it will be a big problem in the coming years. This paper is well written but overlooks some important progress on this topic. Another missing part is the lack of discussion on what causes the model-observation difference. Whether the large-scale environments play an important role is unclear, but should be paid attention in the revision. Except these two, the manuscript is very scientific solid.*

We thank anonymous reviewer 1 for their helpful review and their kind words about our manuscript. Below we respond in detail to the two issues raised.

*1. Page-2 L24-28: Feng et al. (2021) has evaluated mcs simulation in a high-resolution climate models based on a newly developed tracking method by Feng et al. (2019) and Song et al. (2019). Please considering citing these studies here.*

We thank the reviewer for bringing these recent studies to our attention. We have added citations to Feng et al. (2019), Song et al. (2019) and Feng et al. (2021) to our introduction.

*2. How about the role of large-scale environments in the storm initiation in this region? Are the underestimated frequency of storms in the model is caused by the underestimated frequency of large-scale favorable environments in the model? In Feng et al. (2021), it was found that this is the case in the United States Great Plains as large-scale environments are*

*very important for the storm initiation there (Song et al. (2019)). It is good to check whether this is also the case here.*

The reviewer makes a good point about the possible role of large-scale storm environments. However, the purpose of this study was to test whether the use of WRF and TITAN together could produce results that were reasonably realistic when compared to observations, and due to the computational expense required for running the model at convection-resolving resolution the study is restricted to a one-month period. A full evaluation of the role of the large-scale environment is beyond the scope of this paper, and would require simulations over a much larger area and longer time period. To address the reviewer's point, we have updated the conclusions to include a note regarding the possibility that large-scale environments could play a role in the differences we observed, with a reference to Feng et al. (2021).

**Reviewer 2**

*The paper, Object-based analysis of simulated thunderstorms in Switzerland: application and validation of automated thunderstorm tracking on simulation data, is well written and reasonably clear. The content is very interesting, and I agree with the Authors that tracking and lagranian analysis is a fantastic tool in understanding storm evolution and climatology. I applaud the level of technical detail the authors went to in describing the processing chain, this increased the impact of a potential publication as the community has a much better chance to duplicate the methods used and further the science.*

We thank Scott Collis for his very constructive and helpful review.

*There is one minor and one major issue with the manuscript. The minor issue is an incomplete survey of the literature and establishing of why the authors used TITAN when they did not need its most unique feature: dealing with splits and mergers. There has been an increase in the number of object tracking codes and papers that use tracking in studying storms. Notably the TOBAC framework (Heikenfeld et al., 2019) and a recent paper by Fridlind et al. (2019). Since TITAN was designed to work with radar data the authors need to establish why TITAN.*

We are grateful to Dr. Collis for bringing these studies to our attention. We began work on our study before the publication of Heikenfeld et al. (2019) and Fridlind et al. (2019), and chose to use TITAN because it was freely available and well known in the community through its operational use. We have added citations to other tracking methods to our article's introduction, and briefly noted why we chose to use TITAN in Section 2.3.

*The major issue is the authors have not explained why TRT was NOT used to track storms in the model data. The authors are comparing track data from modeled storms using a variety of microphysical schemes derived using TITAN to real storms (I assume gridded using some sort of objective analysis scheme like Barnes, cressman or nearest neighbor) and tracked using TRT. How much of the variation is due to TRT"s method of linking subsequent frames? TITAN has a very sophisticated hessian solver to link object identified in subsequent*

*frames. TITAN and TOBAC have options to also use pre-tracking steps to get close to the neighborhood like FFT image shift and more recently optical flow. This does not seem like an apples vrs apples comparison. How much are the differences between radar and model climatology due to model physics and tracking? This needs to be explained and/or caveated. This is also part of another issue with the paper as it does not go into the "why" enough. Why do we expect different lifecycles of storms due to different microphysical schemes (which is interesting).*

We thank Dr. Collis for this important point. We used TRT for the real storms because we consider it to be the best available object-based representation of radar-observed storms in Switzerland. The algorithm was specifically designed for use in Switzerland with the Swiss radar network, taking into account the unique characteristics of the terrain and hardware used, and has been extensively tested and used operationally.

We were not able to use TRT for the modelled storms because the code for TRT is proprietory to MeteoSwiss and is not available for general use. We therefore chose instead to use TITAN which is freely available. This way we show that an open-source solution to storm tracking compares reasonably well to results from a regionally-specific closed-source method.

There are indeed many possible sources of difference between the modelled and observed storms. We show in the paper that the choice of model microphysics scheme affects the results such that different TITAN thresholds are required for the results to match the TRT observations. Other sources of error include representation of the large scale environment, model resolution and domain, and so on. We have included some more detailed remarks on our choice of TITAN/TRT (Sections 2.2 and 2.3) and possible error sources (Section 4) in the manuscript, and have included a caveat about these differences (Section 4) as suggested.

*One final niggle, the calculation of S-Band reflectivity instead of C-Band. Authors need to establish this is not an issue by invoking smaller drop sizes (always Rayleigh) or such like. This seemed like a very throw-away statement.*

We agree the radar frequency is another source of difference in the comparisons. We used S-band reflectivities from WRF because that is what the various WRF microphysics schemes provide in the `REFL_10CM` output, meaning it is what is easily available to most researchers running WRF. It is apparent from the WRF output that even on the same assumed frequency, the calculated reflectivites differ markedly by microphysics scheme. We see these differences as further reasons for the use of object-based techniques that "abstract away" some of the differences in implementation and attempt to compare core storm properties instead. We have added some clarifying wording to Sections 2.4 and 2.6 and to the conclusions.

**References**

Feng, Z., R. A. Houze, L. R. Leung, F. Song, J. C. Hardin, J. Wang, W. I. Gustafson, and C. R. Homeyer, 2019: Spatiotemporal characteristics and large-scale environments

of mesoscale convective systems east of the Rocky Mountains. **32 (21)**, 7303–7328, doi: 10.1175/JCLI-D-19-0137.1.

Feng, Z., F. Song, K. Sakaguchi, and L. R. Leung, 2021: Evaluation of mesoscale convective systems in climate simulations: Methodological development and results from MPAS-CAM over the United States. **34 (7)**, 2611–2633, doi:10.1175/JCLI-D-20-0136.1.

Fridlind, A. M., and Coauthors, 2019: Use of polarimetric radar measurements to constrain simulated convective cell evolution: a pilot study with Lagrangian tracking. **12 (6)**, 2979–3000, doi:10.5194/amt-12-2979-2019, URL https://amt.copernicus.org/articles/12/2979/2019/.

Heikenfeld, M., P. J. Marinescu, M. Christensen, D. Watson-Parris, F. Senf, S. C. van den Heever, and P. Stier, 2019: tobac 1.2: towards a flexible framework for tracking and analysis of clouds in diverse datasets. **12 (11)**, 4551–4570, doi:10.5194/gmd-12-4551-2019.

Song, F., Z. Feng, L. R. Leung, R. A. Houze Jr, J. Wang, J. Hardin, and C. R. Homeyer, 2019: Contrasting spring and summer large-scale environments associated with mesoscale convective systems over the US Great Plains. **32 (20)**, 6749–6767, doi: 10.1175/JCLI-D-18-0839.1.